

# TSCytoPred: a deep learning framework for inferring cytokine expression trajectories from irregular longitudinal gene expression data to enhance multi-omics analyses

Joung Min Choi[1] and Heejoon Chae[2]

[1] Department of Computer Science, Virginia Polytechnic Institute and State University (Virginia Tech), Blacksburg, VA, United States of America
[2] Division of Computer Science, Sookmyung Women's University, Seoul, Republic of South Korea

Corresponding author
Heejoon Chae,
heechae@sookmyung.ac.kr

## ABSTRACT

Cytokines play a crucial role in immune system regulation, mediating responses from pathogen defense to tissue-damaging inflammation. Excessive cytokine production is implicated in severe conditions such as cancer progression, hemophagocytic lympho-histiocytosis, and severe cases of Coronavirus disease-2019 (COVID-19). Studies have shown that cytokine expression profiles serve as biomarkers for disease severity and mortality prediction, with machine learning (ML) methods increasingly employed for predictive analysis. To improve patient outcome predictions, treatment adaptation, and survival rates, longitudinal analysis of cytokine profiles is essential. Time-series cytokine profiling has been linked to tumor response, overall survival in various cancers, and acute encephalopathy. Similarly, COVID-19 severity and patient outcomes correlate with cytokine expression dynamics over time. However, challenges remain due to the limited availability of time-series cytokine data, restricting broader experimental applications and robust predictive modeling. Recent advancements indicate that cytokine expression can be computationally inferred using gene expression data and transcription factor interactions. Inferring cytokine levels from existing gene expression datasets could enhance early disease detection and treatment response predictions while reducing profiling costs. This work proposes TSCytoPred, a deep learning-based model trained on time-series gene expression data to infer cytokine expression trajectories. TSCytoPred identifies genes relevant for predicting target cytokines through interaction relationships and high correlation. These identified genes are subsequently utilized in a neural network incorporating an interpolation block to estimate cytokine expression trajectories between observed time points. Performance evaluations using a COVID-19 dataset demonstrate that TSCytoPred significantly outperforms baseline regression methods, achieving the highest coefficient of determinataion ($R^2$) and the lowest mean absolute error (MAE). Furthermore, cytokine data inferred by TSCytoPred enhances COVID-19 patient severity risk predictions, demonstrating the model's clinical utility. TSCytoPred can be effectively applied to datasets with limited time points and accommodates longitudinal datasets containing irregular temporal gaps, thereby enhancing disease outcome analysis such as in COVID-19 cases and expanding

the applicability of multi-omics datasets in rare disease contexts with missing multi-omics samples. TSCytoPred is publicly available at https://github.com/joungmin-choi/TSCytoPred.

## INTRODUCTION

Cytokines are intercellular signaling proteins that play a crucial role in nearly every aspect of human immunology (*Jiang et al., 2021*). They activate multiple signal transduction pathways to regulate immune responses, balancing pathogen defense and tissue-damaging inflammation. Cytokine activity is characterized by two key properties: redundancy, where different cytokines exhibit overlapping cellular effects in a specific context, and pleiotropy, where their effects vary depending on cell-type-specific receptor usage (*Ozaki & Leonard, 2002*). A cytokine storm, a phenomenon in which aberrant pathogenic T cells and inflammatory monocytes rapidly produce excessive cytokines, can trigger life-threatening conditions, including cancer progression and hemophagocytic lymphohistiocytosis (*Cabaro et al., 2021*; *Turnquist et al., 2020*; *Zhang et al., 2022*). In recent years, numerous studies have reported that the primary cause of Coronavirus disease-2019 (COVID-19) mortality is a virus-induced cytokine storm, where excessive pro-inflammatory cytokine production leads to acute respiratory distress and widespread tissue damage (*Jiang et al., 2021*; *Cabaro et al., 2021*). Large-scale cohort analyses have shown that elevated serum IL-6, IL-8, and TNF-α are strong predictors of disease severity and mortality (*Del Valle et al., 2020*). Other groups have defined cytokine panels (*e.g.*, IL-6, IL-8, IL-10, IP-10) that stratify patients by outcome and risk (*Cabaro et al., 2021*). These studies provide evidence that cytokine expression profiles serve as predictors of mortality and disease severity in COVID-19. Additionally, recent research has explored the application of machine learning (ML) techniques for the analysis and prediction of COVID-19 outcomes. For example, *Laatifi et al. (2023)* applied explainable ML methods including SHapley Additive exPlanations (SHAP) and Local Interpretable Model-agnostic Explanations (LIME) to plasma cytokine profiles, identifying elevated levels of VEGF-A, MIP-1β, and IL-17 as robust indicators of severe COVID-19, whereas cytokines like RANTES and TNF-α were associated with non-severe cases. Similarly, *Han et al. (2024)* integrated cytokine profiles with electronic health records data with ML algorithms to forecast COVID-19 severity with high accuracy, highlighting the utility of combining immunological and clinical features in predictive models.

Comprehensive analysis of cytokine profiles over time is essential for more accurately predicting patient outcomes, adapting treatments, and improving survival rates. Time-series cytokine profiling offers the potential to identify patterns that signal early disease progression and treatment responses, providing clinicians with actionable insights for personalized patient care based on predicted disease trajectories. Multiple studies

have demonstrated that temporal fluctuations in cytokine levels correlate with local tumor response and overall survival across diverse conditions, including breast cancer, hepatocellular carcinoma, and acute encephalopathy (*Saldajeno et al., 2024*; *Qi et al., 2020*; *Tomioka et al., 2023*). These findings underscore the promise of cytokine trajectory analysis for developing precision medicine approaches that guide therapeutic decision-making and enhance clinical outcomes across heterogeneous patient populations. This temporal approach has demonstrated particular relevance in COVID-19 research, where time-series cytokine expression profiles have been strongly linked to infection outcomes. Through analysis of cytokine level transitions, researchers can predict disease severity, monitor symptom progression, and assess patient prognosis based on dynamic cytokine patterns (*Sanchez-de Prada et al., 2022*; *Deus et al., 2024*).

Existing approaches for modeling cytokine expression over time typically rely on statistical methods such as Auto-Regressive Integrated Moving Average (ARIMA) or recurrent neural network (RNN)-based deep learning models such as long short-term memory (LSTM) networks. For example, *Kavitha et al. (2023)* developed an LSTM-based framework for cardiovascular disease prediction using cytokine profiles, while *Shoaib et al. (2024)* proposed an LSTM classifier to enable targeted coronary artery disease risk stratification using 450 cytokine biomarkers. *Lin et al. (2024)* employed ARIMA and convolutional neural network (CNN) models to predict hepatocellular carcinoma prognosis and identify potential biomarkers. While these methods have shown promise in disease progression and outcome prediction, they face several significant limitations. ARIMA models assume linear relationships between observations and predictions, which may not capture the complex, non-linear dynamics of cytokine interactions (*Fattah et al., 2018*; *Liu, 2024*). Additionally, both ARIMA and traditional RNN approaches typically require regularly spaced time intervals, which is often impractical in clinical settings where sample collection occurs at irregular intervals (*Kontopoulou et al., 2023*; *Mienye, Swart & Obaido, 2024*). RNNs, particularly LSTMs, suffer from vanishing gradient problems, training difficulties, and poor generalization performance on small or sparse datasets—characteristics commonly found in cytokine time-series studies that contain only a few, irregularly spaced measurements (*Noh, 2021*). Transformer and CNN-based models have also been applied for modeling longitudinal cytokine data, but they also often require interpolation to regularize time steps, which can introduce bias (*Lin et al., 2024*; *Yadalam et al., 2025*). Furthermore, a major challenge in this field remains the limited availability of comprehensive cytokine profile datasets, which constrains the development of robust predictive models and hinders the reproducibility of experimental designs necessary for advancing clinical research and improving severity predictions and outcome assessments. These challenges are compounded by irregular sampling intervals, biological variability, and the high cost and invasiveness of cytokine assays. Moreover, cytokine expression data often exhibit high inter-individual variability due to genetic factors. Combined with missing measurements and small sample sizes, these issues complicate modeling efforts and introduce potential biases during preprocessing steps such as normalization and imputation.

Emerging evidence suggests that cytokine expression can be computationally inferred and imputed based on signature gene expression patterns within specific contexts (*Prabahar et al., 2024*). Dysregulation of cytokine expression—driven by transcription factors and genes involved in upstream signaling pathways—has been linked to autoimmune diseases, chronic inflammation, and increased susceptibility to infections (*Turner et al., 2014*). Significant correlations have been identified between gene expression levels and circulating cytokines of the same immune-related genes or proteins (*Young et al., 2020*). In recent years, several databases have been developed to catalog interactions between transcription factors and cytokines, using data from functional assays, *in vivo* binding studies, and *in vitro* DNA binding experiments (*Carrasco Pro et al., 2018*; *Santoso et al., 2020*; *Jiang et al., 2021*). By inferring cytokine profiles from existing time-series gene expression data, researchers can gain deeper insights and make more accurate predictions regarding early disease markers, treatment responses, and disease severity. This approach not only reduces the cost of cytokine profiling but also enhances the integration of multi-omics data for comprehensive disease analysis. Despite its potential, inferring cytokine expression profiles from time-series data presents several challenges. A major limitation is the scarcity of analytical tools specifically tailored for multivariate time-series regression. Current approaches typically rely on machine learning-based multivariate models that treat each time point independently, or on univariate statistical methods; both are limited in their ability to capture complex temporal relationships (*Ospina et al., 2023*). To address this gap, RNN models have recently gained popularity for analyzing time-series data across various applications, including forecasting, prediction, and anomaly detection (*Han et al., 2019*). However, RNN models generally require extensive datasets with numerous time points to deliver robust predictions (*Lipton, Berkowitz & Elkan, 2015*). This requirement poses significant challenges, as publicly available gene expression time-series datasets usually contain only limited time points due to high sampling costs. Furthermore, many longitudinal studies involve samples collected at irregular intervals, resulting in uneven gaps between measurements across patients. Such irregularities complicate the application of traditional RNN models, which inherently assume uniformly spaced time intervals (*Mienye, Swart & Obaido, 2024*).

The present work proposes TSCytoPred, a deep learning-based model for inferring time-series cytokine expression from gene expression profiles. Unlike conventional RNN-based approaches, TSCytoPred employs an RNN-free architecture specifically designed to handle datasets with limited time points and irregular temporal intervals—common characteristics of clinical longitudinal studies. The model incorporates three core modules: (1) a biologically informed gene selection pipeline that integrates transcription factor-cytokine relationships, protein-protein interactions, and correlation filtering to identify the most predictive features, where "strong interaction" refers to protein–protein interactions derived from the STRING (*Szklarczyk et al., 2023*) database with high-confidence combined score and "high correlation" refers to top-ranked Spearman correlations between gene and cytokine expression (see Methods for details); (2) a feed forward multilayer perceptron (MLP) block that extracts complex predictive patterns from gene expression profiles; and (3) an interpolation mechanism that models cytokine

trajectories between observed time points to reconstruct complete temporal profiles. This architecture enables effective utilization of sparse, irregular clinical data while recovering missing cytokine values to enhance downstream applications such as disease severity prediction. Performance evaluation against standard machine learning and deep learning regression models demonstrates that TSCytoPred achieves the lowest mean absolute error (MAE) on real COVID-19 datasets. By generating complete longitudinal cytokine profiles, TSCytoPred significantly improves COVID-19 severity classification performance, demonstrating its clinical utility. The model's ability to accommodate datasets with limited temporal resolution and irregular sampling intervals makes it particularly valuable for disease outcome analysis in clinical settings and expands the applicability of multi-omics approaches in disease research where complete datasets are often unavailable.

The primary aim of this study is to evaluate whether cytokine expression trajectories can be reliably inferred from gene expression data with limited and irregular time points, and whether such inferred trajectories improve downstream clinical outcome prediction. We hypothesize that incorporating biologically informed gene selection together with temporal interpolation will enhance inference accuracy compared with baseline models, and that the inferred cytokine trajectories will provide added value for predicting disease severity.

## METHODS

TSCytoPred comprises two main steps: the gene selection and the time-series inference. The overall workflow is illustrated in Fig. 1 and Supplementary Material S1.

### Data collection and preprocessing

Cytokine and gene expression profiles, along with clinical information from COVID-19 patients, were obtained from the Clinical and Omics Data Archive (CODA) under accession number CODA_D23017 (*Jo et al., 2022*). This dataset contains longitudinal multi-omics data from 300 COVID-19 patients, including single-cell Ribonucleic acid (RNA) sequencing, cytokine profiling, and additional information such as the National Early Warning Score (NEWS) severity score. Additionally, it provides pseudo-bulk RNA sequencing data, which aggregates read counts across different cell types to generate patient-specific gene expression profiles. These aggregated profiles were used as the gene expression dataset in this study.

For analysis, we selected patients who had both cytokine and gene expression data available for three time points within a 15-day period (approximately two weeks). This timeframe was chosen based on reports that COVID-19 symptoms typically last up to two weeks (*Raveendran, 2020*; *Wu, Yu & Lee, 2022*). After filtering, a total of 93 patients met these criteria. Notably, time-series samples were collected at irregular intervals, meaning that the time gaps between successive time points varied across patients. The summaries of clinical severity, based on the distribution of NEWS scores for the 93 patients, and the distribution of time gaps between consecutive timepoints for each patient are provided in Supplementary Material S2.

Data preprocessing followed a similar methodology to previous studies analyzing cytokine profiles associated with COVID-19 disease progression (*Han et al., 2024*). Initially,

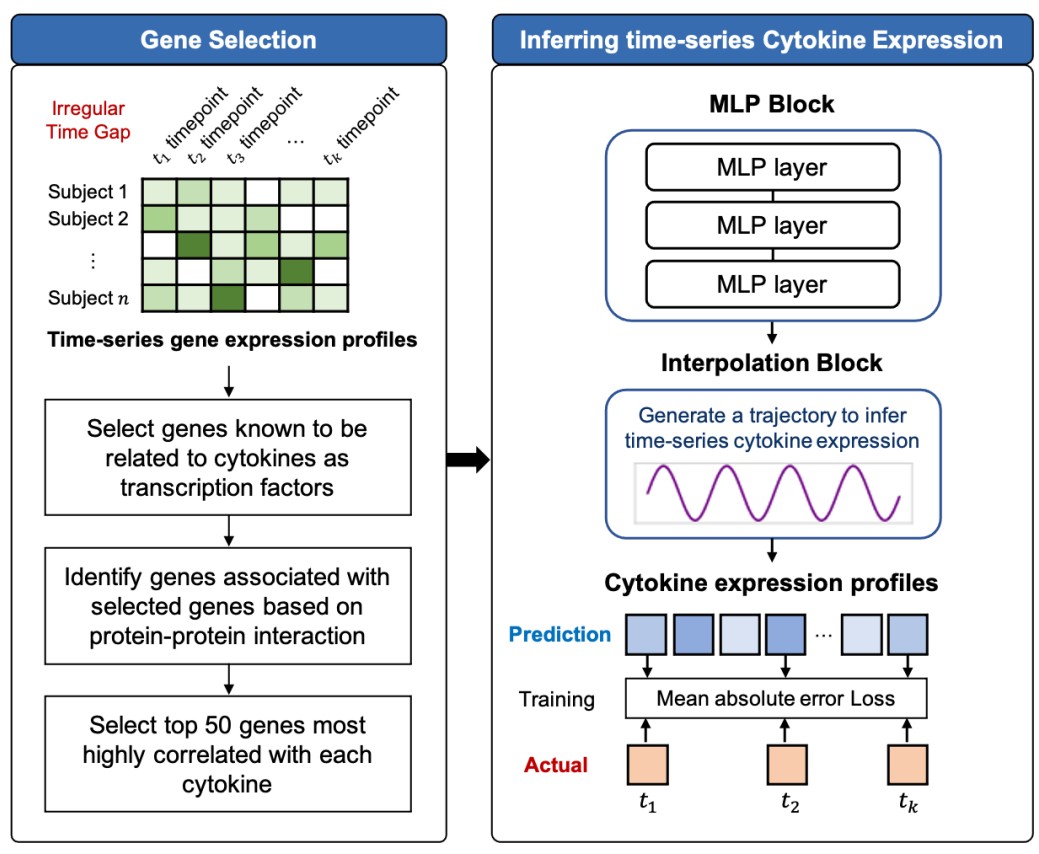

**Figure 1** Workflow of TSCytoPred for inferring the time-series cyotokine expression using time-series gene expression data.

only cytokines and genes shared across all samples were retained. Features with more than 50% missing values were removed to minimize bias, and the remaining missing values were imputed using a random forest imputation approach. To ensure consistency across samples, library size normalization was performed, followed by a log transformation to stabilize variance. After preprocessing, a total of 16,116 genes and 185 cytokines remained in the final dataset for analysis.

## Selecting genes associated with cytokines

To prevent overfitting caused by high dimensionality while effectively capturing interactions between features for cytokine expression inference, the gene selection step follows a three-step process. First, genes that are known to be related to cytokines as transcription factors are selected based on the CytReg database (*Carrasco Pro et al., 2018*; *Santoso et al., 2020*). This database contains an extensive cytokine-gene network derived from functional assays, *in vivo* binding, and *in vitro* DNA binding, extending the existing literature on cytokine interactions. Next, TSCytoPred identifies additional genes associated with the previously selected cytokine-related genes based on protein-protein interactions from the STRING database (*Szklarczyk et al., 2023*). A combined score of 0.9 was used as the threshold for interaction significance. Finally, genes that are highly correlated with cytokines are selected.

The Spearman correlation was calculated between each gene and its corresponding cytokine, and the top 50 genes with the highest correlation were retained. To enhance transparency, we further summarized the strength and significance of these correlations in Supplementary Material S3 (distribution of correlation coefficients across all cytokine–top 50 gene pairs) and Supplementary Material S4 (summary statistics for correlation and corresponding $p$-values including median, interquartile range, minimum, and maximum values for the top 50 genes per cytokine). As a result, our gene selection step identified 3,429 genes, which were then used as inputs for predicting time-series cytokine expression.

## Inferring time-series cytokine expression based on deep learning

TSCytoPred is a deep learning model with an RNN-free architecture designed for multivariate time-series inference (Fig. 1). It is particularly suited for longitudinal datasets with a small number of time points and irregular time gaps. Let $n$ represent the number of genes in the input and $m$ represent the number of cytokines to be inferred. Let $T = (t_1, t_2, \ldots, t_k)$ denote an increasing sequence of times, where $k$ is the total number of time points in the longitudinal data. The gene expression and cytokine expression observations at time $t_i$ are represented as $x_i^g \in \mathbb{R}^n$ and $x_i^c \in \mathbb{R}^m$, respectively. The complete gene expression and cytokine expression matrices are denoted as $x^g = (x_1^g, \ldots, x_k^g) \in \mathbb{R}^{n \times k}$ and $x^c = (x_1^c, \ldots, x_k^c) \in \mathbb{R}^{m \times k}$, respectively. Additionally, the time gap vector, $\delta \in \mathbb{R}^k$, is defined as the time lag between consecutive time points. Specifically, the time gap for the $i$th time point is given by $\delta_i = t_i - t_{i-1}$.

Given the inputs $x^g$, $x^c$, and $\delta$, TSCytoPred first extracts features using the MLP block $l$. This block consists of three fully connected (FC) layers with leaky ReLU activations, which produce the expansion coefficient predictors $\theta_\ell$ as follows:

$$h_{\ell,1} = \text{leakyReLU}(W_{\ell,1} x^g + b_{\ell,1}), \tag{1}$$

$$h_{\ell,2} = \text{leakyReLU}(W_{\ell,2} h_{\ell,1} + b_{\ell,2}), \tag{2}$$

$$\theta_\ell = \text{leakyReLU}(W_{\ell,3} h_{\ell,2} + b_{\ell,3}), \tag{3}$$

where $W_{\ell,j}$ for $j \in 1, 2, 3$ are the weights, and $b_{\ell,j}$ for $j \in 1, 2, 3$ are the bias terms for the MLP block $\ell$. These expansion coefficients are then passed to the temporal interpolation block, which infers the cytokine expression for the given time points while accounting for the time gaps $\delta$ between them. In this framework, "time-series data" refers to longitudinal measurements from the same subject across multiple time points. For each subject, all available time points are processed jointly rather than independently, thereby preserving temporal dependencies across observations. The interpolation block then generates subject-specific cytokine trajectories over time, rather than a single static output, reflecting both the underlying gene expression dynamics and the irregular gaps between samples.

To explicitly address the irregular sampling intervals in our dataset, TSCytoPred incorporates $\delta$ directly into the temporal interpolation block. This design allows the model to use the actual elapsed time between successive samples to guide interpolation, rather than assuming uniformly spaced intervals. By doing so, patients with unevenly collected

measurements are not excluded; instead, the model reconstructs continuous cytokine trajectories that respect the heterogeneous time gaps across individuals.

TSCytoPred employs linear interpolation to generate cytokine expression trajectories between time points $t_1$ and $t_k$, chosen for its computational efficiency and robust performance with sparsely sampled longitudinal data. Linear interpolation has demonstrated comparable or superior predictive performance in time-series tasks while requiring significantly less computational effort than alternatives such as nearest-neighbor methods (*Lepot, Aubin & Clemens, 2017*; *Choi et al., 2023*; *Challu et al., 2023*). To validate this design choice, we developed a TSCytoPred variant using cubic spline interpolation and conducted 5-fold cross-validation experiments. Linear interpolation consistently outperformed cubic spline interpolation across all evaluation metrics (Supplementary Material S5). While TSCytoPred can accommodate higher-degree methods such as polynomial, Hermite, or spline interpolation, these approaches typically require more than four time points, making them impractical for datasets with limited temporal sampling. The interpolation process generates continuous cytokine expression profiles across the entire time interval from $t_1$ to $t_k$, providing cytokine level estimates even at intermediate time points where gene expression data were unavailable (Fig. 1). The interpolation is computed as follows:

$$\hat{x}_i^c = \theta_{\ell,1} + \frac{\delta_i}{t_k - t_i} \cdot (\theta_{\ell,k} - \theta_{\ell,1}), \tag{4}$$

where $\hat{x}_i^c$ denotes the inferred cytokine expression at intermediate time point $t_i$, $\theta_{\ell,1}$ and $\theta_{\ell,k}$ represent the inferred cytokine expression values at initial time $t_1$ and the final time point $t_k$, respectively, and $\delta_i$ represents the elapsed time gap from the initial time point $t_1$.

To train the model, the MAE loss was computed between the actual and inferred cytokine expression values for the non-missing time points. TSCytoPred was trained using the adaptive optimization algorithm Adam, with a learning rate of $10^{-5}$ with batch size of 30, and run for 5,000 epochs. Training was halted if the MAE loss did not improve over the course of 100 consecutive epochs. The model was implemented using the PyTorch (Version 1.6.0) (*Paszke et al., 2019*), Scikit-learn (Version 1.7.1) (*Pedregosa et al., 2011*), Pandas (Version 2.3.1) (*Wes McKinney, 2010*), and Numpy (Version 2.2.6) (*Harris et al., 2020*). Hyperparameters were tuned using grid search, with each experiment repeated five times. The hyperparameters that minimized the average MAE loss were selected as optimal values. The hyperparameter optimization results including the number of hidden nodes, number of layers, learning rate, and batch size are summarized in Supplementary Material S6. The MLP block consisted of three fully connected (FC) layers, with 1024, 512, and a final layer size equal to the number of cytokines to be inferred. The model was evaluated on a single-node server equipped with 40 Intel(R) Xeon(R) Silver 4114 CPU cores running at 2.20 GHz, 250 GB of main memory, and an NVIDIA RTX 8000 GPU with 48 GB of memory.

# RESULTS

## Performance evaluation of TSCytoPred for inferring time-series cytokine expression

TSCytoPred was benchmarked against several widely used regression models, including ML approaches—ElasticNet, Lasso, Ridge, and Linear Regression—as well as the statistical model ARIMA. We further compared TSCytoPred with a neural network-based regression model (NN), which treated each time point as a separate sample and used a similar MLP block-based architecture with the same number of nodes and fully connected layers as in our model. In addition, we evaluated recurrent neural network models, including a LSTM network and a hybrid convolutional neural network using 1D-CNN combined with LSTM (CNN-LSTM), to assess their inference performance. The ML regressors were implemented using 'Scikit-learn' python package. A 5-fold cross-validation was performed at the patient level (*i.e.*, each patient's three time points were kept within the same fold to prevent any data leakage across folds), and the hyperparameters were optimized by grid search based on the average MAE (see Supplementary Material S7). The optimized hyperparameter settings for each regression model were as follows: ElasticNet ($\alpha = 0.1$), Ridge ($\alpha = 10$), Lasso ($\alpha = 0.1$); LSTM (two layers with 2,000 and 1,000 hidden units, respectively); and CNN-LSTM (kernel size = 3, padding = 1, 2,000 channels, followed by two LSTM layers with 1,000 and 250 hidden units). To ensure a fair comparison, the same input data—comprising the genes selected by our gene selection step—was used for all comparison methods. Model performance was evaluated using multiple metrics. MAE and root mean square error (RMSE) quantify the magnitude of prediction errors, with RMSE placing greater weight on larger errors. The $R^2$ coefficient indicates the proportion of variance in cytokine levels explained by the model. Mean absolute percentage error (MAPE) expresses prediction errors as a percentage of actual values, enabling scale-independent comparisons across cytokines. Spearman correlation (CORR) measures the rank-based association between predicted and actual cytokine levels, capturing the consistency of the prediction ordering even when absolute values differ. Evaluations were conducted using multi-omics COVID-19 datasets containing time-series gene expression and cytokine profiles collected from 93 patients across three time points within a 15-day period. A detailed description of this dataset can be found in 'Methods' section.

As summarized in Table 1 (with 95% confidence intervals provided in Supplementary Material S8), TSCytoPred outperformed the other models in cytokine expression inference, achieving the highest average $R^2$ coefficient of 0.257, along with the lowest average MAE (0.437) and MAPE (0.118). ElasticNet achieved the second-best $R^2$ of 0.247, while the NN model had the second-best performance in the other metrics, with an MAE of 0.442 and the best MAPE of 0.116. By contrast, both LSTM and ARIMA produced negative $R^2$ values and higher MAE, indicating that, for datasets with limited and irregularly sampled time points, recurrent models and statistical approaches are disadvantaged due to their reliance on regularly spaced temporal measurements and their need for longer sequences to achieve stable training. Given these poor performances, we excluded LSTM and ARIMA from further experiments. To further evaluate robustness, we also conducted a 3-fold

**Table 1** Average prediction performance results of TSCytoPred with other comparison methods based on the 5-fold cross validation.

| Metric | TSCytoPred | NN | ElasticNet | Lasso | Ridge | Linear | LSTM | CNN-LSTM | ARIMA |
|---|---|---|---|---|---|---|---|---|---|
| $R^2$ | **0.257** | 0.246 | 0.247 | 0.207 | 0.184 | 0.161 | −0.099 | 0.175 | −1.783 |
| MAE | **0.437** | 0.442 | 0.449 | 0.465 | 0.465 | 0.472 | 0.583 | 0.465 | 0.829 |
| RMSE | **0.610** | 0.618 | 0.620 | 0.639 | 0.641 | 0.650 | 0.801 | 0.651 | 1.052 |
| MAPE | 0.118 | **0.116** | 0.125 | 0.131 | 0.123 | 0.125 | 0.171 | 0.121 | 6.6E+04 |
| CORR | **0.986** | 0.985 | 0.985 | 0.984 | 0.984 | 0.984 | 0.983 | 0.989 | 0.366 |

Notes.
Values in bold indicate the best performance among the compared methods.

**Table 2** Average prediction performance results of TSCytoPred and comparison methods based on five-fold cross-validation after removing outlier cytokines: (a) cytokines with outliers in > 1% of samples.

| Metric | TSCytoPred | NN | Linear | Ridge | ElasticNet | Lasso | CNN-LSTM |
|---|---|---|---|---|---|---|---|
| R2 | **0.362** | 0.344 | 0.297 | 0.316 | 0.351 | 0.300 | 0.192 |
| MAE | **0.449** | 0.458 | 0.481 | 0.475 | 0.468 | 0.492 | 0.525 |
| RMSE | 0.605 | 0.617 | 0.614 | 0.606 | **0.592** | 0.619 | 0.704 |
| MAPE | 0.151 | **0.144** | 0.158 | 0.157 | 0.166 | 0.177 | 0.178 |
| CORR | **0.992** | 0.991 | 0.990 | 0.990 | 0.991 | 0.990 | 0.988 |

Notes.
Values in bold indicate the best performance among the compared methods.

cross-validation, which yielded results consistent with the 5-fold setting, confirming that TSCytoPred's performance remains stable under different partitioning schemes (Supplementary Material S9). Furthermore, TSCytoPred demonstrated greater stability in performance across the 5-fold cross-validation, exhibiting less variation compared to the other methods. We recognize, however, that the overall $R^2$ values remain relatively modest, suggesting limited variance explained. To investigate this further, we conducted sensitivity analyses by removing cytokines with varying proportions of outlier samples (defined using interquartile range (IQR)-based thresholds of >5%, >3%, and >1%). These additional results (Table 2, Supplementary Material S10) showed that after removing outlier cytokines, the average $R^2$ of TSCytoPred improved from 0.257 to 0.362, while most comparison methods also achieved $R^2$ values above 0.3. Importantly, TSCytoPred remained the best-performing method across all scenarios, supporting the robustness of the model even when noisy cytokines were excluded.

To further assess robustness, we also repeated the evaluation using a larger subset of COVID-19 patients with two time points within 9 days (125 patients), and TSCytoPred continued to outperform other models in most metrics (Supplementary Material S11). In addition, we tested the model on the original three–time point cohort by predicting cytokine abundance dynamics in reverse, where TSCytoPred again outperformed baseline methods, including NN and CNN–LSTM (Supplementary Material S12). These results confirm that TSCytoPred provides reliable inference performance even when longer-term cytokine trajectories are unavailable or when different temporal dynamics are considered.
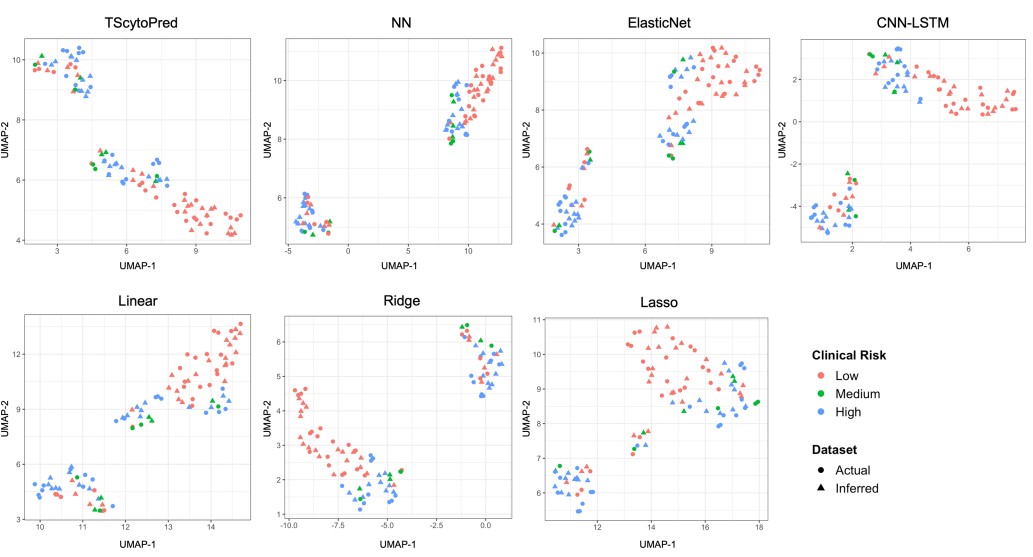

**Figure 2** UMAP visualization of actual and inferred cytokine expression profiles in test data for COVID-19 dataset using the proposed and comparsion methods, colored by clinical risk and shaped by actual and inferred dataset.

We also assessed model performance by comparing actual and inferred cytokine expression values through visualization. For this purpose, the dataset was randomly split into an 8:2 ratio for training and testing solely for visual evaluation, with models trained on the training set. This split was not used for reported performance metrics; instead, all quantitative evaluations in this study were conducted using five-fold cross-validation to ensure robustness and reproducibility. We first visualized the cytokine expression profiles using uniform manifold approximation and projection (UMAP) at the sample level to assess how closely regression models infer to actual data, categorized by clinical COVID-19 risk (Fig. 2). Ideally, inferred profiles (triangles) should cluster closely with actual measured profiles (circles) within each clinical risk category. TScytoPred exhibited the strongest alignment, as indicated by the close proximity of inferred to actual data points, especially notable within the medium-risk category (green). In contrast, methods such as Lasso displayed significant discrepancies, with inferred points broadly dispersed from actual measurements. Similarly, Ridge and Linear methods showed noticeable separations between inferred and actual profiles, reflecting lower accuracy in inference.

To further evaluate the performance, we examined the individual cytokine predictions. The predicted cytokine expression values in the test data were compared to the actual values using the Wilcoxon signed-rank test. All $p$-values from these tests were adjusted for multiple comparisons using the false discovery rate (FDR) correction (Benjamini–Hochberg procedure). Among the 185 cytokines, 84.32% (156 cytokines) of those inferred by TSCytoPred showed no significant difference from the actual values (p-value > 0.05) as shown in Table 3. In contrast, other regression methods exhibited significant discrepancies between predicted and actual cytokine expression, with the following numbers of cytokines showing no significant differences: 152 (NN), 149 (ElasticNet), 155 (Linear), 147 (Lasso),

**Table 3** Number (and percentage) of cytokines inferred by each regression method without statistically significant difference from actual values ($p > 0.05$).

| # of cytokines | TSCytoPred | NN | ElasticNet | Linear | Lasso | Ridge | CNN-LSTM |
|---|---|---|---|---|---|---|---|
| 185 | 156 (84.32%) | 152 (82.16%) | 149 (80.54%) | 155 (83.78%) | 147 (79.46%) | 153 (82.70%) | 80 (43.24%) |

153 (Ridge) and and 80 (CNN-LSTM). As an example, Supplementary Material S13 presents boxplots comparing the distributions of actual and inferred cytokine expression for a few key cytokines associated with COVID-19. TSCytoPred exhibited the closest alignment to the actual expression distribution, effectively capturing the patterns observed in the real data. This demonstrates that TSCytoPred outperforms the other methods in inferring individual cytokine patterns.

## Effectiveness of gene selection in TSCytoPred

TSCytoPred employs a comprehensive gene selection strategy incorporating three key steps to identify gene sets associated with cytokines, facilitating accurate inference of cytokine expression. To assess the effectiveness of each step, we developed three variants of TSCytoPred by sequentially modifying or removing components of the gene selection procedure. Specifically, the 'CytReg' variant selects only transcription factor genes directly associated with cytokines based on the CytReg database. The 'CytReg + STRING' variant further includes additional genes connected to these cytokines through protein-protein interactions from the STRING database. Finally, the complete approach, labeled 'CytReg + STRING + CORR', represents the full TSCytoPred method, where we evaluated performance using varying numbers of top-ranked genes according to Spearman correlation with each cytokine (*e.g.*, 'top 10' refers to selecting the 10 genes with the highest correlation per cytokine). Additionally, we tested another variant utilizing mutual information (MI)—a common alternative feature selection method in time-series analysis—in place of Spearman correlation. Performance across all variants was evaluated using 5-fold cross-validation.

The comparative results, summarized in Table 4 (with 95% confidence intervals provided in Supplementary Material S14), demonstrate that the full TSCytoPred approach, integrating genes selected from CytReg, STRING, and Spearman correlation rankings, achieved the highest predictive accuracy, with an average $R^2$ of 0.257 and MAE of 0.437. In contrast, replacing correlation-based selection with mutual information did not yield significant improvements, despite using a comparable number of genes. These findings underline the effectiveness and importance of our gene selection strategy for enhancing cytokine expression inference from gene expression data.

## Improvement of the severity prediction through cytokine expression inference using TSCytoPred

Cytokine levels have been widely recognized as predictors of mortality and illness severity in COVID-19 infections (*Del Valle et al., 2020*; *Cabaro et al., 2021*; *Onuk et al., 2023*). In this study, we explored whether utilizing cytokine data inferred through TSCytoPred could enhance severity outcome prediction, specifically using the NEWS. The NEWS system assesses the severity of illness based on seven criteria: respiratory rate, oxygen saturation,

**Table 4** Average prediction performance of TSCytoPred under different feature sets based on the 5-fold cross validation.

| Metric | CytReg | CytReg + STRING | CytReg + STRING + CORR | | | | CytReg + STRING + MI | | | |
|---|---|---|---|---|---|---|---|---|---|---|
| | – | – | Top 10 | Top 20 | Top 30 | Top 50 | Top 10 | Top 20 | Top 30 | Top 50 |
| # of genes | 242 | 875 | 1,569 | 2,141 | 2,628 | 3,429 | 2,086 | 3,131 | 4,021 | 5,615 |
| $R^2$ | 0.097 | 0.134 | 0.238 | 0.244 | 0.245 | **0.257** | 0.162 | 0.193 | 0.195 | 0.201 |
| MAE | 0.487 | 0.476 | 0.444 | 0.442 | 0.443 | **0.437** | 0.463 | 0.456 | 0.456 | 0.454 |
| RMSE | 0.676 | 0.659 | 0.621 | 0.618 | 0.619 | **0.610** | 0.642 | 0.634 | 0.633 | 0.632 |
| MAPE | 0.128 | 0.125 | 0.115 | 0.116 | 0.116 | **0.117** | 0.120 | 0.120 | 0.119 | 0.120 |
| CORR | 0.988 | 0.989 | 0.990 | 0.990 | 0.990 | **0.990** | 0.990 | 0.990 | 0.990 | 0.990 |

Notes.
Values in bold indicate the best performance among the compared methods.

temperature, blood pressure, and heart rate (*Martín-Rodríguez et al., 2022*). The score ranges from 0 to 21, with three points awarded per criterion, where higher scores indicate more abnormal values and a more severe condition. In our COVID-19 dataset, clinical information for the NEWS severity score was available for 113 patients. Among them, 93 had both gene expression and cytokine data at three time points, while the remaining 20 had only gene expression data. A detailed description of the dataset can be found in the 'Methods' section.

To investigate whether inferred cytokine data could enhance the prediction of disease severity outcomes, we compared regression models trained with and without inferred cytokine profiles. We employed an ElasticNet regressor—selected based on its robust performance in previous evaluations—and trained it using available cytokine data along with inferred cytokine expression values for 20 patients who originally had only gene expression data. Cytokine data inferred by TSCytoPred and other baseline methods were evaluated and compared against predictions from a regressor trained solely on existing cytokine measurements (*i.e.,* without inferred cytokine data).

The average $R^2$ and MAE from 5-fold cross-validation showed that training the regressor with the inferred cytokine profiles from TSCytoPred led to a 6.12% performance improvement, with the average $R^2$ increasing from 0.294 to 0.312 (Fig. 3). In contrast, other methods showed little improvement, with the NN model achieving the second-best $R^2$ of 0.297. These findings suggest that TSCytoPred could help the regression model to effectively leverages inferred cytokine data to enhance the prediction of clinical severity outcomes, especially for patients lacking direct cytokine measurements.

## Enhancing clinical risk prediction for COVID-19 severity outcome using cytokine profiles

While predicting severity scores for each timepoint is important, directly predicting a patient's overall clinical risk based on cytokine profiles could offer a more efficient approach for practical applications. Ultimately, patients will be classified into clinical risk categories based on the NEWS score threshold, and their responses will be predicted accordingly. In this experiment, we transformed the NEWS severity score values into categorical outcomes based on the predefined thresholds used in hospitals (*Welch, Dean &*

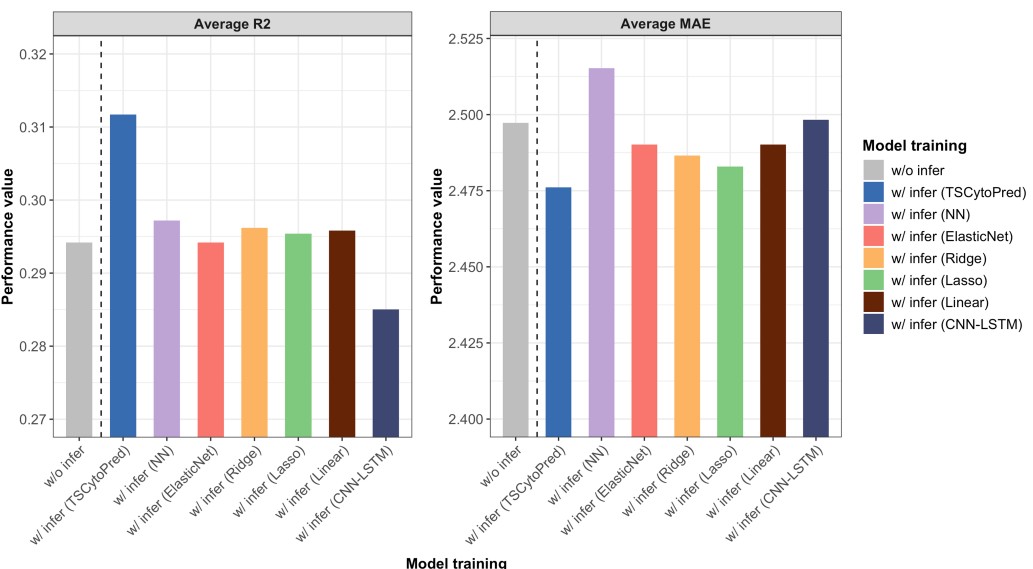

**Figure 3** Average performance results for the NEWS severity score predictions of the regressor trained with the addition of the 20 inferred subjects using different methods, repeating 5-fold cross validations.

**Table 5** Pre-defined clinical risk system classifying based on the NEWS score threshold.

| NEWS score | Clinical risk | Response |
|---|---|---|
| Aggregate score 0-4 | Low | Ward-based response |
| Aggregate score 5-6 | Medium | Key threshold for urgent response |
| Aggregate score 7 or more | High | Urgent or emergency response |

*Hartin, 2022*), as shown in Table 5. A support vector machine (SVM) classifier was selected and constructed using the 'Scikit-learn' package with default parameters due to its superior classification performance compared to methods including decision trees and Naive Bayes (*Zhang et al., 2017*), taking cytokine expression profiles as input.

The performance of the classifier for clinical risk prediction was assessed by training the models with cytokine data inferred by TSCytoPred for the 20 patients lacking direct cytokine measurements, along with data inferred by other baseline methods. Model performance was evaluated using F1-scores based on 5-fold cross-validation, under two distinct prediction scenarios. In the first scenario (time-point-level prediction), models aimed to predict clinical risk at individual time points. In the second scenario (patient-worst-level prediction), the goal was to predict the highest clinical risk level that a patient might experience throughout the entire course of illness. Predicting the worst-case clinical risk is especially valuable for clinicians, as it allows them to anticipate patient needs and implement targeted interventions. For example, if a patient exhibited clinical risk levels of 'Low,' 'Medium,' and 'Low' across three different time points, the task would be to correctly identify 'Medium' as the highest risk level experienced by that patient.

**Table 6 Average F1-score results for the clinical risk predictions of the classifier trained with the addition of the cytokine profiles of 20 inferred subjects using different methods, based on 5-fold cross validations.**

| Testing scenario | w/o infer | w/ infer | | | | | | |
|---|---|---|---|---|---|---|---|---|
| | | TSCytoPred | NN | Linear | ElasticNet | Lasso | Ridge | CNN-LSTM |
| Timepoint-level | 0.565 | **0.602** | 0.570 | 0.596 | 0.540 | 0.547 | 0.562 | 0.573 |
| Patient-worst-level | 0.568 | **0.592** | 0.565 | 0.508 | 0.526 | 0.550 | 0.528 | 0.527 |

**Notes.**

Values in bold indicate the best performance among the compared methods.

As summarized in Table 6 (with 95% confidence intervals provided in Supplementary Material S15), the addition of inferred cytokine samples from TSCytoPred improved the average F1-score in both scenarios. For timepoint-level prediction, the average F1-score improved from 0.565 to 0.602, while for patient-level prediction, it improved from 0.568 to 0.592. TSCytoPred outperformed all other methods in both severity score and clinical risk predictions. These results demonstrate that TSCytoPred is an effective tool for inferring missing cytokine profiles that would otherwise be discarded, and that the inclusion of inferred data significantly enhances model performance.

## DISCUSSION AND CONCLUSIONS

In this study, we introduced TSCytoPred, a multivariate time-series inference model designed to predict cytokine expression from gene expression data. The model identifies key genes involved in the interaction between gene expression and cytokines, leveraging transcription factor relationships, protein-protein interactions (PPI), and gene-cytokine correlations. These selected genes are used as input for the inference, which features an MLP-based block to extract relevant features. These features are then passed to an interpolation block that generates a trajectory, allowing the model to predict corresponding cytokine expression values. In this work, we employed linear interpolation as the strategy for reconstructing cytokine trajectories. While various higher-degree methods (*e.g.*, polynomial, cubic Hermite, spline) could in principle be applied, such approaches typically require more than four time points and are therefore less suitable for longitudinal datasets with sparse temporal sampling. We found linear interpolation to be more stable and accurate under these conditions, as confirmed in our supplementary comparison with spline interpolation. Evaluations using a COVID-19 dataset, including 5-fold cross-validation, statistical testing, and visualization, demonstrated that TSCytoPred outperforms baseline methods. It achieved the highest average $R^2$ and the lowest average MAE, RMSE, and MAPE. These improvements, while modest in absolute numerical terms, are particularly meaningful in the context of noisy and incomplete longitudinal cytokine datasets. Even small gains in $R^2$ and error reduction can enhance the stability of inferred trajectories, leading to more reliable downstream tasks such as disease severity stratification and outcome prediction. The effectiveness of the gene selection strategy was also assessed against various gene sets, confirming its ability to enhance the accuracy of cytokine expression predictions. Furthermore, the study explored the improvement of disease severity prediction through both regressors and classifiers, utilizing the incomplete

longitudinal COVID-19 dataset. The results indicated that TSCytoPred effectively supports disease severity prediction, demonstrating its potential to aid in clinical applications.

One limitation of TSCytoPred is that it was tested exclusively on the COVID-19 dataset due to challenges in finding a multivariate time-series dataset that includes both cytokine and gene expression profiles with a sufficient number of samples for training and evaluation. Since TSCytoPred was developed and evaluated exclusively in the context of COVID-19, caution is warranted in extending our findings to other diseases. Cytokine dynamics may differ substantially across conditions such as autoimmune disorders, cancer, or other infectious diseases, and further testing will be necessary to determine the model's generalizability. In particular, future studies applying TSCytoPred to more heterogeneous datasets—such as those with larger numbers of timepoints, more diverse patient populations, or entirely different cytokine profiles—will be essential to rigorously validate its robustness across disease contexts and to demonstrate its adaptability to varying biological settings. Nevertheless, the design of TSCytoPred offers flexibility that could support applications beyond COVID-19. For example, in datasets where cytokine–gene associations are weaker or sample sizes are smaller, the feature selection module can be tuned to prioritize high-confidence interactions while employing regularization to reduce noise. Similarly, for studies spanning longer follow-up periods, the interpolation block is capable of handling irregular time gaps, allowing the model to infer trajectories in scenarios where longitudinal data collection is sparse or uneven. These features make TSCytoPred adaptable to a broad range of clinical and biological contexts, provided further validation is performed.

Another important limitation is the lack of experimental validation to confirm the biological relevance of the inferred cytokine values. While our computational analyses provide supportive evidence—for example, UMAP visualizations showing that inferred cytokine expression profiles cluster closely with actual measurements within clinical COVID-19 risk categories—wet-lab or clinical validation will be essential to further establish the biological interpretability of the inferred immune trajectories.

To provide additional biological context for our gene selection, we performed functional enrichment analysis of the 3,429 genes selected by our pipeline using DAVID (*Sherman et al., 2022*) (Supplementary Material S16). The top enriched pathways included Cytokine–cytokine receptor interaction, PI3K–Akt signaling pathway, and COVID-19, all central to immune regulation and cytokine dynamics. Notably, genes such as CCL2 and IL6R were among those highly selected, consistent with their known roles as mediators of hyperinflammatory responses and cytokine storm in COVID-19 (*Qudus et al., 2023*; *Chakraborty et al., 2020*). These findings lend biological credibility to our gene selection strategy and further support the relevance of TSCytoPred in modeling cytokine dynamics.

To the best of our knowledge, this is the first model to apply deep learning for inferring multivariate time-series cytokine expression from gene expression data. By adopting an RNN-free architecture, TSCytoPred is well-suited for time-series datasets with a limited number of timepoints and can handle longitudinal data with irregular time gaps. While TSCytoPred was specifically designed for cytokine datasets, the approach could be extended to other omics data types. For instance, the model could be trained to infer time-series

methylation data. Similar to gene selection for cytokines, genes related to CpG sites within the same promoter region could be identified and fed into the inference step. However, as TSCytoPred is optimized for cytokine data, it is important to tailor and fine-tune the model for specific omics data types to achieve optimal performance.

We believe that our model will significantly support cytokine research by enabling the use of time-series gene expression data and enhancing the analysis of incomplete longitudinal multi-omics datasets. In particular, TSCytoPred facilitates the integration of transcriptomic data with cytokine profiles, effectively imputing one omics layer from another. This can be especially valuable in longitudinal multi-omics studies where certain data types, such as cytokines, are missing or difficult to obtain. This can improve the accuracy of disease outcome predictions and contribute to a better understanding of disease mechanisms. From a clinical perspective, inferring cytokine trajectories from gene expression may help identify patients at risk of hyper-inflammation or cytokine storm at earlier stages and support outcome prediction when direct cytokine measurements are not available. In practice, inferred trajectories could complement real-time diagnostics by providing an early-warning proxy of immune dysregulation, enabling clinicians to stratify patients and personalize interventions such as immunosuppressive or anti-cytokine therapies. Moreover, compared with extensive experimental cytokine profiling—which can be costly, invasive, and logistically difficult in routine clinical workflows—our computational framework offers a more cost-effective and scalable alternative that leverages transcriptomic data already being collected. We anticipate that TSCytoPred can contribute to improved patient management and a deeper insight into the temporal behavior of the immune system. Future work will focus on validating inferred cytokine trajectories against experimental cytokine profiling to confirm their biological relevance.

## ACKNOWLEDGEMENTS

We acknowledge the use of ChatGPT to improve the grammar and the writing style of the manuscript.

### Funding

This research was supported by the "Korea National Institute of Health" (KNIH) research project (project No. 2024-ER-0801-01) and by the Bio&Medical Technology Development Program of the National Research Foundation (NRF) funded by the Korean government (MSIT) (No. RS-2025-18732993). The funders had no role in study design, data collection and analysis, decision to publish, or preparation of the manuscript.

### Grant Disclosures

The following grant information was disclosed by the authors:
The "Korea National Institute of Health" (KNIH) research project: project No. 2024-ER-0801-01.

The Bio&Medical Technology Development Program of the National Research Foundation (NRF) funded by the Korean government (MSIT): No. RS-2025-18732993.

## Competing Interests

The authors declare there are no competing interests.

## Author Contributions

- Joung Min Choi conceived and designed the experiments, performed the experiments, analyzed the data, prepared figures and/or tables, authored or reviewed drafts of the article, and approved the final draft.
- Heejoon Chae conceived and designed the experiments, authored or reviewed drafts of the article, and approved the final draft.

## Data Availability

The cytokine and gene expression profiles, and clinical information from COVID-19 patients, are publicly available from the Clinical and Omics Data Archive (CODA): CODA_D23017.

The code and data are available at Github and Zenodo:

- https://github.com/joungmin-choi/TSCytoPred
- Joung Min Choi. (2025). joungmin-choi/TSCytoPred: v0.0.03 (v0.0.3). Zenodo. https://doi.org/10.5281/zenodo.15345163.

## Supplemental Information

Supplemental information for this article can be found online at http://dx.doi.org/10.7717/peerj.20270#supplemental-information.

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
