# Peer review of "TSCytoPred: a deep learning framework for inferring cytokine expression trajectories from irregular longitudinal gene expression data to enhance multi-omics analyses"

_PeerJ, doi:10.7717/peerj.20270_

## Round 0.1 · original submission · Major Revisions

· Academic Editor

Major Revisions

We got three in-depth reviews, which highlight numerous aspects of the manuscript for reconsideration. I find them constructive and virtually helpful, so I ask the Authors to prepare a revised version of the manuscript in a detailed response to the reviewers' comments.

I find the manuscript interesting and the presented method potentially useful. I hope the revision will improve the reporting for the benefit of the Readership.

Reviewer 1 ·

Basic reporting

Title and Abstract
While the title is clear, it could better emphasize the potential clinical applications or real-world implications of TSCytoPred (e.g., predictions of COVID-19 severity and outcomes).
Consider adding a phrase to highlight the primary application or significance of the work, such as "for predicting disease severity" or "to enhance patient outcome predictions." For example:
"TSCytoPred: A Deep Learning Framework for Inferring Cytokine Expression Trajectories and Predicting Disease Severity from Irregular Longitudinal Gene Expression Data."
• The abstract lacks a strong statement about the clinical relevance and impact of the study's findings. While it mentions COVID-19 as a use case, it does not sufficiently emphasize how TSCytoPred can address real-world issues like predicting patient outcomes or advancing multi-omics analysis.
• The abstract focuses heavily on technical aspects but provides insufficient context regarding the broader scientific or healthcare-related contribution, potentially limiting appeal to a general scientific audience.
• Some jargon (e.g., "MAE," "R²") may not be instantly clear to readers unfamiliar with the methodology, and the abstract does not define these terms.
Suggestions for Improvement:
• Add a line addressing the clinical or practical implications.
For example: "The model’s ability to infer missing cytokine data and enhance severity predictions positions it as a valuable tool for advancing disease outcome analysis, such as in COVID-19 cases."
• Simplify technical language where possible, ensuring terms like "MAE" or "R²" are accompanied by context or interpretation.
• Consider rephrasing the final sentence to call out the broader utility of TSCytoPred explicitly.

Introduction
The Introduction of the manuscript provides an adequate foundation for understanding the research, but some areas could benefit from further elaboration to strengthen the background and contextual framing.
While the introduction briefly mentions machine learning methods for cytokine analysis, it does not adequately discuss specific prior works or current state-of-the-art methods. This makes it harder to assess how the proposed model substantially advances the field.
• What are the current methods for addressing irregular longitudinal cytokine data?
• Why are RNNs insufficient? Are there prior efforts to replace RNNs for this type of data analysis?
• While the introduction frames the importance of analyzing cytokine expression trajectories, it does not adequately connect this to the broader implications for clinical research and patient management.
• The challenges of working with deep learning models on small or irregular datasets are not explored in sufficient depth. This is key to contextualizing the novelty and importance of the TSCytoPred architecture.
For example, why are traditional deep learning models like RNNs or existing time-series algorithms (e.g., Long Short-Term Memory [LSTM]) poorly suited to handle these challenges?
• The introduction does not address the inherent biological variability in cytokine expression data or the potential for bias from imputation techniques.
• Discussion about the preprocessing concerns or the biological noise found in longitudinal gene expression and cytokine datasets could provide a more comprehensive understanding for the reader.
• The model-specific novelty (e.g., RNN-free architecture, interpolation block) is barely touched upon in the introduction, with little to no detail about how these features uniquely address the research gap.
• This creates a gap between the identified problem and the proposed solution’s justification.
Suggestions for Improvement:
• Include a brief survey of existing approaches to cytokine profiling and time-series data modeling (including RNN-based methods) to provide a stronger context for the knowledge gap.
Discuss challenges or successes of existing multivariate regression models, RNNs, and machine learning approaches used in cytokine analysis.
• Deepen the discussion of the clinical implications of cytokine trajectory predictions.
• Example: Elaborate on how TSCytoPred could improve patient care, early disease detection, progression monitoring, or therapy decisions, particularly for diseases like COVID-19 or cancer.
• Include a paragraph discussing why traditional machine learning or deep learning methods (e.g., RNNs, LSTMs) fail to handle irregular time-series datasets effectively.
• Example: Highlight their limitations in terms of data requirements, model complexity, or inability to interpolate between uneven time gaps.
• Provide more detail upfront about why TSCytoPred is innovative and how it addresses the identified challenges (e.g., its interpolation block, gene selection, and RNN-free architecture).
• Briefly address challenges inherent to working with cytokine and gene expression datasets:
o Biological noise and variability
o Missing data and imputation concerns, and
o Small sample sizes in longitudinal studies.
• Add more detail about the use of the COVID-19 dataset to help the reader understand why it’s an appropriate application for testing TSCytoPred. Mention specific aspects of cytokine behavior in COVID-19 (e.g., their role in cytokine storms, correlation with severity outcomes).
Suggested Outline for Improvement:
Emphasize their role in disease and potential for clinical applications. Discuss the scarcity of time-series datasets, irregular timepoints, and limitations of current modeling methods (e.g., RNNs). Briefly describe machine learning approaches for cytokine expression prediction and highlight their deficiencies. Connect the challenges to the missed opportunities for disease monitoring and prediction. Provide a more robust description of the model’s innovative architecture (gene selection, interpolation block, RNN-free structure) and explain how it uniquely addresses the identified gaps. Emphasize real-world applications, particularly regarding COVID-19 and other use cases.

Figures & Tables
The figures and tables are suitable. They do not require any further revisions.

Experimental design

Materials and Methods
While the methodology is well-structured and sufficiently detailed in parts, there are gaps in reproducibility, validation, and statistical design that need addressing. Expanding the dataset size, providing more specific parameter details, and validating the biological plausibility of inferred trajectories will significantly strengthen the study.
However, there are some limitations related to clarity and detail:
• Important dataset preprocessing steps (e.g., handling missing data via random forest imputation and variance-stabilizing log transformations) are described but lack detail on parameters and procedures. For instance:
o How was the random forest imputation validated for effectiveness?
o Were outliers examined and addressed in the cytokine/gene expression data?
• Details about selecting patients and filtering datasets could be expanded. For instance, why were the 15 days chosen for longitudinal samples? Was it guided by clinical considerations or statistical design?
• While some training details (e.g., layer sizes, optimization) are covered, there is no detailed pseudocode or hyperparameter tuning explanation (e.g., choice of "3 hidden layers" or dropout rate, if any). Hyperparameter optimization details should be more explicitly referenced, especially given the potential variability in neural network results.
• The specific choice of linear interpolation as a temporal bridging method is justified by computational simplicity, but alternative methods (e.g., neural ODEs, spline interpolation) are not fully explored or evaluated. This could hinder reproducibility in scenarios with different time gaps or datasets.
• Although highly relevant databases (CytReg and STRING) are utilized, the inclusion threshold (e.g., correlation thresholds for gene selection) is heuristic. Was this threshold empirically validated to ensure optimal performance? The manuscript could provide justification or sensitivity analyses regarding these cutoffs.
• A diagram of the data flow (e.g., from patient selection to cytokine trajectory inference) would greatly improve clarity for readers replicating the work.
• The general methodology is outlined clearly, including preprocessing, gene selection, and the TSCytoPred architecture. Parameters such as batch size and number of epochs can be inferred from the training details.
• Precise information about the Clinical and Omics Data Archive (CODA) dataset is missing. For example:
o How were the selected 93 patients distributed in terms of clinical severity? Were there potential biases in the dataset that could affect inference?
o Were confounding variables (e.g., demographics, comorbidities) considered?
• There is no description of how time gaps or irregular intervals were distributed in the dataset, which could significantly impact generalization to other datasets.
• Details on data splits (e.g., 8:2 train-test ratio) are present but vague regarding strategies like cross-validation, which is critical for reproducibility.
• With five-fold cross-validation stated in the results, more details on how splitting was performed to mitigate potential overfitting or data leakage are recommended.
The 15-day time window selected for longitudinal sampling is partially justified based on previous findings that COVID-19 symptoms typically resolve in two weeks. However, there are limitations to this choice:
• Cytokine dynamics often exhibit trajectories spanning weeks or months (e.g., in chronic diseases). Limiting the timeframe may underrepresent longer-term cytokine fluctuations that could affect the accuracy of the model in predicting severe outcomes.
• Since only three time points are collected, the model may lack sufficient resolution to fully capture cytokine dynamics, especially for patients with irregular and uneven time intervals.
• The dataset includes longitudinal data from only 93 patients, which may limit the generalizability of the findings. For a deep learning model, this is a relatively low sample size.
• The small size increases the risk of overfitting, despite gene selection steps to reduce dimensionality.
• The manuscript relies heavily on MAE, RMSE, and R² as evaluation metrics but offers no justification for why these were selected as the primary metrics.
• Explaining these choices about the primary goal (accurate imputation of missing cytokine points and severity prediction) would strengthen the justification.
• The choice of statistical methods (e.g., linear interpolation) was not compared to other potential approaches (e.g., advanced regression models, neural ODEs).
• For imputation and inference of irregular multidimensional time-series data, newer methods (e.g., temporal convolutional networks, self-supervising architectures) could offer improvements.
• Beyond statistical alignment, no validation is presented to confirm that inferred cytokine values are biologically relevant or consistent for interpreting immune trajectories
• While TSCytoPred addresses COVID-19 cytokine prediction, it is unclear whether the findings will translate to other diseases or time-series cytokine analyses. This should be explicitly discussed as a limitation.
• There is no biological or experimental validation to confirm that inferred cytokine trajectories represent actual cytokine dynamics. Experimental validation would strengthen confidence in the model’s predictions.
• The small sample size and limited time points restrict the generalizability of findings. Additional datasets (e.g., longitudinal cytokine profiles from cancer or autoimmune diseases) should be tested to demonstrate the robustness of TSCytoPred.

Validity of the findings

Results
The results of the study, presented in the paper "TSCytoPred: A Deep Learning Framework for Inferring Cytokine Expression Trajectories from Irregular Longitudinal Gene Expression Data," demonstrate meaningful contributions to the field, primarily in advancing computational techniques for analyzing irregular and limited long-term cytokine datasets. However, there are certain limitations, which are detailed below:
• TSCytoPred's performance is tested exclusively on a single dataset (COVID-19 cytokine profiles), which limits its generalizability to other diseases or datasets. For example, cytokine dynamics in cancer, inflammatory diseases, or autoimmune disorders might differ significantly.
• Using additional longitudinal omics datasets would strengthen the applicability and robustness of the reported results.
• While TSCytoPred demonstrates superior performance over baseline methods, the improvements in metrics are modest (e.g., only a 4% better R² relative to ElasticNet). This raises questions about whether the added architectural complexity justifies the gain, particularly in practical use cases.
• The study lacks experimental validation of inferred cytokine trajectories. Although statistical results support the plausibility of the model's predictions, there is no direct evidence demonstrating that predicted values align with actual biological cytokine dynamics.
• The clinical application (COVID-19 severity prediction) is narrow and does not sufficiently demonstrate the broader utility of the model (e.g., in other diseases or multi-omics analyses). Expanding use cases would showcase its adaptability and potential for widespread adoption.
Concerns regarding the plausibility and credibility of the results presented in the paper:
• The study uses data from only 93 COVID-19 patients with three time points each, which raises concerns about statistical power and generalizability. While the cross-validation approach mitigates overfitting, testing on additional datasets is necessary to confirm the results' reproducibility.
• The random forest imputation step for missing values introduces further variability, yet no sensitivity analysis (to assess its impact) is presented.
• The dataset’s irregular time gaps are discussed in the methods but not rigorously analyzed in the results. It would be valuable to evaluate how well TSCytoPred performs under various degrees of irregularity, as this aspect is a core feature of the model.
• Although the results show improvement over baseline models, the R² value of 0.257 remains relatively low. This suggests a limited explanation of variability in cytokine dynamics, which could point to noise in the dataset or model limitations. Further refinement or more informative features might be needed to improve explanatory power.
• While the model selects genes using CytReg, STRING, and correlation thresholds, it does not provide biological insights into why the 3429 genes selected are particularly relevant to cytokine dynamics. A discussion of key pathways or functional modules would add credibility to the approach.
Suggestions for Improvement:
• Test TSCytoPred on additional datasets (e.g., chronic inflammatory diseases, cancer cytokine profiles) to demonstrate its broader applicability.
• Incorporate biological validation by comparing inferred cytokine trajectories to independent experimental measurements.
• Assess the impact of different gene selection thresholds, random forest imputation strategies, or degree of irregularity in time gaps on the model's performance.
• Where possible, include additional COVID-19 patient data or pool data from other publicly available cytokine studies to improve statistical power.
• Provide more detail on how inferred cytokine dynamics could be used in decision-making or treatment planning.
• Include functional annotation of the top-selected genes to explain their biological relevance to cytokine dynamics or immune processes.

Discussion
The findings in the discussion are largely consistent with the presented results and align well with the study’s objectives, demonstrating significant progress in cytokine trajectory modeling. However, the discussion would benefit from stronger connections to biological insights, broader disease applications, and a deeper reflection on limitations and potential improvements. Presenting a clearer narrative about the model’s implications in the context of immunology, clinical outcomes, and multi-omics research would significantly enhance its relevance and impact in the field.
The findings are moderately aligned with the broader research context but could be more explicitly connected to key questions in omics and immunology. For example:
• The study establishes a novel approach for irregular longitudinal modelling without RNNs, addressing computational challenges in this domain.
• However, while the discussion mentions the model’s applicability to other cytokine datasets, the contextual relevance to multi-omics or personalized medicine is underexplored. The discussion could elaborate on:
o How TSCytoPred could advance monitoring of other conditions like cancer, autoimmune diseases, or chronic inflammatory conditions.
o How the model could integrate with other data types (e.g., proteomics or metabolomics) for better predictive analysis.
• The focus on COVID-19 is timely, but the relevance of the findings of the paper to other disease areas is not adequately discussed. For broader impact:
o The authors could contextualize the model’s flexibility for future datasets with different biological or clinical parameters.
o Key questions include: Would the model perform similarly in datasets with smaller cytokine-gene associations or longer study timeframes?
• While the discussion briefly mentions generalizability, there is inadequate emphasis on validating the model across other diseases or datasets. The results focus heavily on one dataset, and the discussion does not adequately speculate on how TSCytoPred might perform in more heterogeneous datasets (e.g., datasets with more time points or entirely different cytokine profiles).
• The discussion does not effectively link the statistically derived results to the biological meaning of cytokine trajectories, such as:
o Are certain gene-cytokine interactions highlighted by the model biologically plausible?
o How do the selected genes contribute to understanding cytokine dynamics in immune regulation?
• The study compares TSCytoPred with baseline regression models and an MLP architecture, but the discussion does not address how it compares with other more advanced time-series models (e.g., transformers or temporal convolutional neural networks). More discussion on such comparisons or acknowledging these limitations would enhance the rigor of the findings.
• The discussion underplays the potential impact on clinical decision-making. For instance:
o How could inferred cytokine trajectories complement real-time clinical diagnostics?
o Could the model assist with personalized treatment strategies (e.g., targeting cytokine storms)?
o What are the cost-benefit implications of using such techniques versus increasing experimental cytokine profiling?
Suggestions:
• Provide stronger links between the selected cytokine-gene interactions and their relevance to immune pathways.
• Consider discussing key cytokines highlighted in the dataset (e.g., COVID-19 cytokines involved in cytokine storms) and their biological significance.
• Acknowledge explicitly that TSCytoPred has not been validated across diverse datasets or diseases. Discuss plans for future work to validate it on broader datasets.
• Expand on how TSCytoPred could integrate into multi-omics frameworks.
• Highlight its utility in diseases beyond COVID-19 and its potential to guide therapies or resource allocation in hospitals through inferred cytokine trajectories.
• Provide more explicit comparisons with other advanced time-series models.
• Discuss TSCytoPred’s novelty in deeper technical detail, especially its ability to handle irregular sampling effectively.

Conclusion
The conclusions moderately align with the findings obtained in the study but overemphasize the technical aspects while underexploring biological interpretation, broader applications, and clinical relevance. While the results provide strong evidence of TSCytoPred's utility, the conclusions could better expand on generalizability, biological impact, and its potential for transforming longitudinal omics research and clinical practice. By addressing the following suggestions, the study could present a more impactful, well-rounded narrative, emphasizing both technical strength and translational relevance.
While the results and discussion highlight the use of CytReg and STRING databases to identify biologically relevant gene-cytokine relationships, the conclusions mostly focus on technical aspects of the model. There is no mention in the conclusions about the biological interpretation or relevance of the findings, such as:
o Which cytokines or gene interactions contributed most to the predictive power of TSCytoPred?
o How does the inferred cytokine data align with known immune responses, particularly in COVID-19?
• The conclusions fail to sufficiently address the potential generalizability of TSCytoPred to other datasets, diseases, or experimental contexts.
• Though the findings demonstrate efficacy in COVID-19 severity prediction, the conclusions do not expand on how the model could be utilized for other applications, such as cancer, autoimmune diseases, or chronic inflammatory conditions. This omission limits the impact of the study's broader utility to disease modelling.
• The conclusions highlight improved predictive power (e.g., R², MAE) but do not offer much interpretation of the magnitude of improvements. While performance metrics are presented clearly in the results, the conclusions do not discuss why these metrics are meaningful or how they reflect real-world improvements in cytokine trajectory prediction.
• Although clinical implications are mentioned (e.g., enhancing disease severity predictions), they are not elaborated upon enough. For example:
o How could inferred cytokine trajectories be incorporated into clinical workflows or decision-making tools?
o What are the practical advantages of computational inference versus experimental cytokine profiling in terms of cost, time, or accessibility?
• While the conclusions briefly acknowledge that TSCytoPred has not yet been validated on non-COVID-19 datasets, this limitation needs stronger elaboration regarding how the study will ensure its generalizability. For example:
o What types of datasets (e.g., cancer, chronic inflammation) could reasonably benefit from TSCytoPred?
o Are there specific conditions or dataset characteristics (e.g., small sample size or irregular time points) where the methodology is expected to fail?
• The conclusions make no mention of biological validation for inferred cytokine trajectories, leaving a gap in justifying the credibility of the predictions. Future work could be mentioned here to validate predicted trajectories with actual experimental data.
• While the results demonstrate the model's effectiveness, the conclusions should broaden the scope to include its potential utility in other omics fields beyond cytokine profiling (e.g., methylation, proteomics). This could highlight its broader contribution to computational biology.
• The R² value achieved by TSCytoPred (0.257) is still relatively low, despite outperforming baseline models. The limitations of this result should be more explicitly acknowledged in the conclusions. Additionally, its relevance to real-world interpretation of immune responses (or clinical utility) should be clarified.
Suggestions:
• Summarize key biological insights obtained from the model. For example:
o Did predicted cytokine trajectories align with known immunological dynamics in COVID-19?
o Were certain gene-cytokine interactions particularly critical in improving predictions?
• Discuss the potential adaptability of TSCytoPred to other diseases or omics datasets, emphasizing its utility across different contexts.
• Provide a stronger focus on how the findings translate into clinical impact, including how TSCytoPred could improve patient outcomes, reduce costs, or inform treatment strategies

·

Basic reporting

The authors only mention RNN layers but not LSTM, which should work much better than classic RNN layers. Is there a reason for not trying LSTM-type networks? The authors mentioned that they have a small number of time points, but it would be nice to see the results with LSTM or RNN.
Since the dataset used for training is small (N=93) and Machine Learning models often require big data, is there an alternative (statistical method) to compare with?

“Notably, time-series samples were collected at irregular intervals, meaning that the time gaps between successive time points varied across patients.”
How was this solved?

“A combined score of 0.9 was used as the threshold for interaction significance. Finally, genes that are highly correlated with cytokines are selected. The Spearman correlation was calculated between each gene and its corresponding cytokine, and the top 50 genes with the highest correlation were retained.”
This analysis requires a supplementary table or figure showing p-values, correlation coefficients, where the cutoff was

Since you are doing many tests, the p-value has to be corrected using appropriate p-value correction methods (FDR, Bonferroni, etc).

“Various interpolation strategies can be used, but higher-degree methods such as polynomial, cubic Hermite, or spline interpolation typically require more than four time points, which may not be feasible for longitudinal datasets with fewer time points.” Which one was used in the method? This sentence should be used in the introduction or discussion, not in methods, where authors have to be exact regarding which methodology they used.

“Linear interpolation has demonstrated comparable or superior predictive performance in time-series tasks while requiring significantly less computational effort compared to alternatives such as nearest-neighbor methods.” Is this tested for your dataset? Supplementary figure may be

“The MLP block consisted of three fully connected (FC) layers, with 1000, 500…”
The number of layers and layer sizes should be computationally evaluated in order to find the best combination. I would suggest grid search or the Optuna framework.

The 5-fold cross-validation might slice the data into too small chunks for the model to effectively train. I would suggest using a 3-fold CV or hold-out CV.

Please add X and Y axis annotations on Figure 2 (UMAP-1 and UMAP-2, for example, representing two new UMAP features)

Please explain in more detail how you identified genes with strong interactions and high correlations with cytokines (lines 93-94). How is "strong interaction” defined?

Please provide the information about the computer configuration that you used in your research.

Please provide the versions of the Python software packages that you used in your implementation.

The expression “time-series data” and “time-series models” is used in the paper without defining whether the observations are processed independently or together for a single subject. The “time series” assumes taking dependent time-series data of a single subject and processing it as such. It is unclear to me when authors are processing time-dependent inputs together and producing a single output, versus producing time-dependent outputs.

The paper requires more figures, which will explain this process.

General and optional advice is to look into the cGAS-STING pathway and already developed biomarkers for cGAS-STING activation.

I see a batch size of 30 was used for modeling. Is there a reason why 30 was used? Some values should be tested and compared to see the optimal batch size.

Experimental design

Some sentences in the methods section could fit better in the Introduction or the discussion.

Validity of the findings

The authors provided code and example data to replicate the results. The code and entire setup are clear.

Reviewer 3 ·

Basic reporting

1. The manuscript’s overall objectives are stated (investigating cytokine dynamics), but the specific aims and hypotheses are not clearly delineated in the Introduction. Please explicitly list the primary research questions and the hypotheses tested.

2. "The necessity of time series analysis" is asserted without citation or concrete rationale. The authors should (a) summarize existing approaches to analyzing longitudinal cytokine data, (b) highlight their limitations, and (c) explain why a dedicated time-series framework would overcome these gaps.

3. Further literature review is needed — although the authors emphasize the importance and popularity of the topic, they reference only a few studies.

4. Typo at Line 38 - Cytokine are -> Cytokines are

Experimental design

While the manuscript presents an original deep learning framework and is within the scope of the journal, the research questions are not explicitly formulated, nor are the hypotheses clearly stated about the gaps in the current modeling of cytokine trajectories. The experimental rationale is underdeveloped: the authors should clarify why their model is preferable over existing time-series inference approaches (e.g., RNNs, spline models) and whether the model’s assumptions align with the characteristics of the dataset. Although the Methods section contains sufficient technical detail to replicate the model, the design lacks justification for choices such as linear interpolation and the exact number of genes used in training. The authors should provide justification for these choices, citing relevant references or presenting ablation results.

In terms of performance assessment, their use of 5-fold cross-validation is concerning. As outlined in https://ploomber.io/blog/nested-cv/, this approach can lead to overly optimistic results, especially when model selection is involved. Nested cross-validation would have been more appropriate.

Validity of the findings

The results are promising in terms of predictive accuracy, but the validity of the findings is undermined by several limitations that are insufficiently acknowledged.

1. The model is evaluated solely on a single dataset, with only three irregularly spaced timepoints per patient. This raises concerns about overfitting, generalizability, and statistical power, especially given the model’s complexity.

2. The absence of external validation or transfer to other biological datasets makes it difficult to assess robustness beyond the specific case studied.

3. More critically, no biological interpretation is provided: the model identifies predictive genes but does not investigate their roles in cytokine regulation or disease mechanisms. Without such context, the findings lack translational relevance.

Additional comments

Regarding Code and Reproducibility:

1. The repository should include a requirements.txt (or pyproject.toml/Pipfile) to pin and install dependencies with a single command.

2. Provide a setup.py or modern pyproject.toml and publish the package to PyPI (e.g., tscytopred) so users can install via pip install tscytopred.

3. Integrate pytest with a minimal suite of tests and configure CI (such as GitHub Actions) to run tests on every push or pull request.

4. Running bash run_TSCytoPred.sh currently raises:
NameError: name 'build_time_series_dataset' is not defined. Did you mean: 'build_test_time_series_dataset'?

5. 
Please correct the function call or add the missing definition, and include a short example in the README or a demo notebook to demonstrate end-to-end execution without errors.

---

## Round 0.2 · accepted · Accept

· Academic Editor

Accept

Thank you for submitting revised version of the manuscript. All Reviewers are satisfied with the changes made. Hence, I'm pleased to accept the text for publication. Please mind minor suggestions by the Reviewer 3 - possible to be corrected on the proofs.

Reviewer 1 ·

Basic reporting

Title & Abstract
The authors have revised the title as suggested and satisfactorily addressed all suggestions in the abstract by defining the abbreviations and incorporating the translational/clinical relevance of their algorithm. The title and abstract look good now.

Introduction
The authors adequately addressed all the suggestions for the introduction by adding the background on cytokine roles and their clinical utility; challenges in time-series cytokine data: irregularity, noise, small sample size; limitations of traditional statistical methods and RNNs; a clear articulation of how TSCytoPred uniquely addresses the existing gaps and the potential for real-world applications, especially COVID-19 severity prediction.

Figures & Tables
The figures and tables require no modifications.

Experimental design

Material and Methods
The authors have satisfactorily addressed all the suggestions for the methodology. All concerns are addressed and clarified by the authors with additional data added in the Supplementary Materials.

Results
The authors have adequately addressed all the suggestions. All concerns are addressed and clarified by the authors with additional analysis. The authors have addressed the limitations justly, and the work opens a new way of research.

Validity of the findings

Discussion
The authors have addressed the suggestions and concerns in the discussion. With the addition of new references, the authors have effectively contextualized their findings, addressing concerns about generalizability and the limitations of the model.

Conclusion
It is good to see that the authors have incorporated the suggestions properly and bridged the gap in the conclusion. The conclusion section looks good.

Reviewer 3 ·

Basic reporting

The authors have revised all the points.

Experimental design

The authors have revised all the points.

Validity of the findings

The authors have revised all the points.

Additional comments

* Line 93 - J et al., 2018 needs to be corrected.
* The reference format is inconsistent, should be unified.